# Red Horse Chestnut and Horse Chestnut Flowers and Leaves: A Potential and Powerful Source of Polyphenols with High Antioxidant Capacity

**DOI:** 10.3390/molecules27072279

**Published:** 2022-03-31

**Authors:** Agnieszka Monika Bielarska, Jakub Wojciech Jasek, Renata Kazimierczak, Ewelina Hallmann

**Affiliations:** 1Warsaw Department of Burns, Plastic and Reconstructive Surgery, Military Institute of Medicine, Szaserów 128, 04-141 Warsaw, Poland; abielarska@wim.mil.pl; 2Warsaw University of Medicine, ul. Zwirki i Wigury 61, 02-091 Warsaw, Poland; jakub.jasek07@gmail.com; 3Department of Functional and Organic Food, Institute of Human Nutrition Sciences, Warsaw University of Life Sciences, Nowoursynowska 159c, 02-776 Warsaw, Poland; renata_kazimierczak@sggw.edu.pl

**Keywords:** red horse chestnut, flowers, leaves, polyphenols, anthocyanins, carotenoids

## Abstract

*Aesculus* flowers and leaves are an excellent source of bioactive compounds, including flavanols, phenolic acids, and anthocyanins, and the leaves also contain antioxidant carotenoids and chlorophylls. The aim of this study was to analyse and compare the amounts of bioactive compounds present in *Aesculus hippocastanum* and *Aesculus* × *carnea* flowers and leaves over two years. These two species from six independent locations (parks and green areas) located in Warsaw were assessed in this study. The dry matter by the scale method and polyphenol, carotenoid, and chlorophyll content by the HPLC method of the flowers and leaves was evaluated. Red horse chestnut flowers contained significantly more total carotenoids (40.6 µg/g FW) and chlorophylls (36.9 µg/g FW) than horse chestnut flowers, and red horse chestnut flowers contained higher levels of anthocyanins (5.41 µg/g FW) than other species. We observed that horse chestnut flowers were characterized by a higher total polyphenols concentration (9.45 µg/g FW) compared to red horse chestnut flowers. In addition, the analysis of leaves showed that all quality parameters were higher in red horse chestnut species. Five individual anthocyanins were identified in both species’ flowers, but a higher concentration was found in red horse chestnut flowers, and pelargonidin-3-O-glucoside was the predominant form among a pool of total anthocyanins. In both experimental years, leaves (109.25 mMol/100 g FW and 112.0 mMol/100 g FW) were characterized by a higher antioxidant activity than flowers (27.0 mMol/100 g FW and 27.5 mMol/100 g FW).

## 1. Introduction

Polyphenols are common among plant secondary metabolites [1,2,3]. Plants produce polyphenols as a response to biotic and abiotic environmental stress conditions [4]. In addition, however, polyphenols are commonly called “natural pesticides” [5]. Pest attacks stimulate plants to produce and concentrate secondary metabolites, mostly phenolic acids and flavonoids. Plants develop different defence mechanisms, aided by biochemical bioactive compounds [6]. Unfortunately, plants cannot synthesize defence antibodies as animals do via their immune system. Therefore, plants evolved to produce numerous antimicrobial substances called phytoalexins. Many of these are polyphenols [7,8]. UV radiation is another environmental trigger for polyphenol production. In many dark flowers, a higher concentration of anthocyanins occurs as a protection from intense sunlight [9]. In other plant parts, including leaves, UV radiation stimulates flavonol synthesis [10]. Horse chestnut and red horse chestnut are two of the best-known ornamental and medicinal tree species. The decorative effects of trees are visible, especially in springtime when they flower. Flowers in spectacular inflorescences reach up to 20 cm in height. *Aesculus hippocastanum* is characterized by white flowers with light red spots, but *Aesculus* (×) *carnea* flowers are red. The presence of polyphenols in the leaves, seeds, flowers, and bark of horse chestnut has been reported in the literature but not in red horse chestnut because it is a less well-known species. According to the literature, the content of polyphenols is: in horse chestnuts leaves (7.81–24.48 mg/g FW) [11], seeds (24.24–70.40 mg/g FW) [12], flowers (8.17 mg/g) [13], and bark (363.58 mg/g FW) [14]. For medicinal purposes, fruits, seeds, and cortex are used. Extracts and concentrates are used for creams and ointments with antiinflammatory properties for use in treating phlebitis and different kinds of cancer. However, no culinary properties of chestnut or horse chestnut flowers have been reported. If vegetables and fruits are a useful source of bioactive compounds in our diet, flowers can also be used. There are many plants with edible flowers. Their use in foods could expand the culinary arts, as an old adage suggests: “What is healthy for your stomach should be nice for eyes as well”. However, even though it may not be possible to create a complete diet from leaves and flowers rich in bioactive compounds such as polyphenols, fruits, vegetables, and flower parts are good sources of polyphenols in our diet. Many experiments have shown that flowers can be consumed and used in infusions and tinctures [15,16]. Previous research with black and bristly locust flowers showed that flowers of this species are a reliable source of flavonoids and phenolic acids [17]. Only a limited number of experiments on polyphenol content in the flowers and leaves of red horse chestnut and horse chestnut have been reported. The aim of the present work was to assess the polyphenol content and antioxidant capacity in the flowers and leaves of red horse chestnut and horse chestnut over two annual growth cycles

## 2. Results

The flower and leaf dry matter content in both species is presented in Table 1. In the first year of the experiment, the flowers contained less dry matter than the leaves. In the second year, the flowers of both species contained comparatively more dry matter than in the first year, but the values between years were not statistically significant (*p* > 0.05). Red horse chestnut was characterized by a higher dry matter content in leaves in both years of the experiment but not the flowers, although the results were not statistically significant. Leaves of red and horse chestnut are a much better source of total carotenoids than the flowers. Carotenoids are connected mostly with chlorophyll content. Red horse chestnut leaves were characterized by a higher concentration of total carotenoids than white leaves (*p* < 0.0001). A comparable situation was noted with the flowers, but only in the first year (2018). In the case of chlorophylls, we observed that the two species varied significantly in the content of this pigment in both leaves and flowers (Table 1).

We observed that in 2018 the total polyphenol content in the flowers of both species was similar. In 2019, both species contained more total polyphenols in the leaves than in the flowers, but in 2018, the situation was the opposite. Horse chestnut flowers and leaves were characterized by a higher concentration of total phenolic acids than those of red horse chestnut. It is worth noting that in 2019 the amounts of polyphenols were much higher than those in 2018 and the difference was statistically significant (*p* < 0.0001). However, we did not observe significant differences in the total flavonoid content between the flowers of the two species. However, in both years, the leaves contained significantly more (*p* < 0.0001) total flavonoids than the flowers. Purple colorants (anthocyanins) were detected only in red horse chestnut flowers. The leaves of both species are anthocyanin-free. Detailed carotenoid and chlorophyll analysis showed three individual carotenoids as well as two chlorophylls in both the flowers and leaves of both species (Figure 1).

In both years, we observed a higher concentration of lutein in flowers than leaves. Moreover, red horse chestnut contained more lutein than horse chestnut and the results were statistically significant. The zeaxanthin content was higher in leaves than in the flowers in both years for both species. Red horse chestnuts contained significantly more zeaxanthin (*p* < 0.0001) in both leaves and flowers than horse chestnuts in both years (Table 2).

The main carotenoid characterized for both species is beta-carotene. In 2018 and 2019, red horse chestnuts contained significantly more (*p* < 0.0001) of this carotenoid than horse chestnut leaves and flowers. In both years, red horse chestnut contained significantly more chlorophyll a (*p* < 0.0001) in leaves than horse chestnuts. Although chlorophyll a was detected in the flowers of both species, red horse chestnut characteristically contained a higher concentration than occurred in chestnut (Table 2). Chlorophyll b was more abundant in red horse chestnut. Both flowers (*p* < 0.0001) and leaves (*p* < 0.0001) of red horse chestnut contained significantly more chlorophyll b than horse chestnut in 2018 and 2019. Six different anthocyanin compounds were identified in both species but only in flowers, not leaves (Table 3). In five of the six cases, red horse chestnut contained significantly more anthocyanins than horse chestnut. This is because of the intensive pink flower colour in the red horse chestnut (Figure 2).

Cyanidin-3-O-glucoside occurred in only small amounts in horse chestnut flowers but was three times higher in red horse chestnut flowers in both years, and the differences were statistically significant (*p*-value < 0.0001). The highest concentration among anthocyanin pigments was of pelargonidin-3-O-glucoside. No differences between species were observed in delphinidin-3-O-glucoside content. Both species in both years contained similar amounts of anthocyanin. Malvidin-3-O-glucoside was the only anthocyanin that occurred at a higher concentration in horse chestnut flowers, measuring 0.54 mg/g in 2018 FW and 0.38 mg/g in 2019 FW, while red horse chestnut flowers contained only 0.25 mg/g (2018) FW and 0.26 mg/g (2019) FW, and the results were statistically significant (*p* = 0.0061). Peonidin-3-O-glucoside occurred at a higher concentration in red horse chestnut flowers in both years, while horse chestnut flowers contained 0.67 mg/g (2018) FW and 0.40 mg/g (2019) FW of peonidin-3-O-glucoside (Table 3).

The last anthocyanin pigment identified in the flowers of both species was petunin-3-O-glucoside with red horse chestnuts containing significantly more (*p* < 0.0001) than occurred in horse chestnut.

We identified five phenolic acids and seven flavonoids in the flowers of both species (Figure 3). The dominant phenolic compound was gallic acid of which horse chestnut leaves and flowers contained significantly more (*p* < 0.0001) in both years compared to red horse chestnut leaves and flowers (Table 4). Chlorogenic acid was detected at higher concentrations in horse chestnut and the results were statistically significant (*p* = 0.0072). There were no statistically significant differences between the leaves and flowers of both species in 2018 and 2019. In both years, caffeic acids were more abundant in red horse chestnut flowers (*p* < 0.0001) but higher in the leaves of horse chestnut (*p* < 0.0001). In both years, the flowers of horse chestnuts contained significantly more p-coumaric acid than those of red horse chestnut, while in leaves the higher concentration was in red horse chestnut. We observed that both the flowers and leaves of horse chestnuts contained significantly more ferulic acid in both years than red horse chestnut (*p* < 0.0001). Conversely, in both years we found higher concentrations of quercetin-3-O-rutinoside in red horse chestnut flowers and leaves than in horse chestnut flowers and leaves (*p* < 0.0001). The flowers of the horse chestnut contained significantly more kaempferol-3-O-glucoside than flowers of the red horse chestnut (*p* < 0.0001), while this was reversed in the leaves in which a greater amount was found in red horse chestnut leaves than horse chestnut leaves (Table 4).

The results showed that the flowers of both red and horse chestnut flowers contained significantly more quercetin-3-O-glucoside than the leaves, measuring at 0.39 mg/g FW in 2018 and 0.28 mg/g FW in 2019, respectively. In the leaves, the concentrations of quercetin-3-O-glucoside were 0.26 mg/g FW and 0.19 mg/g FW. The flowers of horse chestnuts contained significantly more quercetin-3-O-glucoside (*p* = 0.0052), but in the case of red horse chestnuts, it occurred more in the leaves than the flowers. In both years, red horse chestnuts contained significantly more myricetin (*p* = 0.0016) than horse chestnut. Luteolin was one of the phenolic compounds with a higher concentration in flowers and leaves, just after gallic acid. In 2018, the flowers and leaves of horse chestnut were characterized by a higher concentration of these phenolic compounds. This trend was not observed in 2019. The concentration of quercetin was significantly different only in the examined plant organs. In both years, we observed that leaves contained more (*p* = 0.0003) than flowers (Table 4). There were no differences between species in either year. Red horse chestnut contained more kaempferol in flowers, while horse chestnut contained more kaempferol in leaves. This situation was observed in both years. We observed that leaves were characterized by a higher total antioxidant activity than flowers in both years. In 2018, red horse chestnut had a higher total antioxidant status than horse chestnut (*p* < 0.0001). However, in 2019, a slightly higher antioxidant status was observed in the horse chestnut species.

## 3. Discussion

In the present study, the species with a white flower colour contained more dry matter compared to those with red flowers. The dry matter content in plants was genetically diverse. The range of dry matter content in different *Aesculus* species was 11.0–17.3 g/100 g FW. This value is similar to those presented by others for different species with white- and pink-coloured flowers [17]. The obtained results are the opposite to those presented in the literature. Pink and red flowers contained more dry matter compared to white flowers [18]. Carotenoids are bioactive compounds that play a key role as pro-health compounds. In the present study, the leaves of both species contained a high concentration of carotenoids. The obtained value of 723.9 µg/g FW for horse chestnut leaves and 923.5 µg/g for red horse chestnuts leaves is similar to popular leafy vegetables such as green lettuce (727.2 µg/g FW) and much higher than that in spinach (125.4 µg/g FW) [19,20]. Healthy properties of leaves can provide chlorophylls as well polyphenols. According to the literature, chlorophylls are molecules that play a vital role as chelators of heavy metals [21,22]. In our study, the concentration of chlorophylls in horse chestnut and red horse chestnut leaves was much higher than that of leafy vegetables such as ice-head lettuce (52 mg/100 g FW), butter-head lettuce (41 mg/100 g FW), and spinach (112 mg/100 g FW). For horse chestnuts, it was 118.5 mg/100 g FW, and for red horse chestnuts, it was 170.5 mg/100 g FW [23]. Polyphenols with a high antioxidant status play a significant role in maintaining good health [24,25]. Many diseases, such as cancer and cardiovascular diseases, are associated with low antioxidant status and polyphenol intake with foods such as fruits and vegetables [26,27]. Flowers and leaves are potentially good sources of dietary polyphenol compounds [28]. The health effects of eating flowers have been examined for many years [29,30,31,32]. In the case of horse chestnuts and red horse chestnuts, any therapeutic effects have mostly been connected and examined with fruits and seeds, not flowers and leaves. In our experiment, we showed that the flowers and leaves of both species are a useful source of polyphenol compounds. To approximate the concentrations of polyphenolic compounds in the flowers and leaves of horse chestnuts, the obtained values should be compared with those of fruits and vegetables. One of the important fruits recommended as a good source of polyphenols is apples. The range of total polyphenol concentrations in five cultivars was shown to be 148–190 mg/100 g FW [33]. In our experiment, we showed that horse chestnut flowers contained total polyphenols in a range of 445–945 mg/100 g FW, much higher than occurs in apples. Citrus fruits are considered to be among the best sources of flavonoids. Different citrus fruits contain total flavonoids in the range of 14.2–70.6 mg/100 g FW [34]. However, our experiment focused on the flowers of horse chestnuts, which are a much better source of total flavonoids with a range of these compounds in flowers of 249–398 mg/100 g FW. Berries are a group of fruits with a high anthocyanin content. Flowers of red horse chestnuts contain a satisfactory level of anthocyanin compounds compared with berry fruits such as highbush blueberry, strawberry, and raspberry [35]. In addition to diminishing disease risk, other functions of plant flavonoids connected with horse chestnut flowers, fruits, leaves, and seeds include vessel strength and wound healing [36]. Therefore, it may be worth combining the health-promoting and therapeutic effects of horse chestnut flowers and leaves due to their high content of polyphenolic compounds, particularly flavonoids. Skin disorders including wounds are common indispositions. Various risk factors accompany wound healing, such as microbial infection and inflammation with high ROS (reactive oxygen species) generation. Applying food with a high concentration of bioactive compounds can reduce these problems [37]. Determining the content of antioxidant agents in edible flowers is one of the most important aims of such experiments. In our experiment, we showed that both red horse chestnuts and horse chestnuts are excellent sources for bioactive compound content. Flowers can be used not only for decorative dish purposes but also as food. A previous experiment with black and bristly locust flowers showed that not only typical edible flowers, such as pansy, rose or daisy flowers, can be used for consumption purposes [17]. The concentration of bioactive compounds in flowers is connected with their colour. The main colorant of flowers is anthocyanins, which are characterized by a high antioxidant status. In our experiment, as found in previous experiments, we show that colourful flowers contain more phenolic compounds than similar species with white flowers. Red horse chestnut flowers with a pink colour also contained more kaempferol and quercetin-3-O-glucoside than a closely related horse chestnut species with white flowers (Table 4). Comparable results were obtained with black and bristly locust flowers and daisy flowers [17,18].

## 4. Materials and Methods

### 4.1. Flowers’ and Leaves’ Origin

Our experiment focused on two Aesculus species: *Aesculus hippocastanum* (horse chestnut) and *Aesculus* (×) *carnea* (red horse chestnut). The first is characterized by white flowers with small red spots (Figure 4), and the second species has pink flowers (Figure 5). Our experiment was conducted over two years, 2018–2019. Flowers of both species were collected in the morning from six independent trees located in Warsaw parks and green areas between 1 and 4 of May each year and the samples were quickly transported to the laboratory. The location of trees was: Lazienki Park (52°22″ N; 21°03″ E), Saski Park (52°14″ N; 21°0″ E), Królikarnia Park (52°41″ N; 21°42″ E), Mokotowskie Pole Park (52°21″ N; 20°99″ E), Sowiński Park (52°23″ N; 20°95″ E), Szczęśliwicki Park (52°21″ N; 20°96″ E).

### 4.2. Plant Material Preparation

Inflorescences, similar to leaves of both species, were gently divided into single flowers. From each tree, 200–300 individual flowers and 50 leaves were collected. Each tree was treated as a single replication. The fresh weight of the samples was between 250 and 350 g per tree for flowers and 150–170 g per tree for leaves. Each species samples were divided into two parts. The first part was used for dry matter evaluation, and the second part was freeze-dried using a Labconco (2.5) freeze-dryer (Warsaw, Poland, −40 °C, pressure 0.100 mbar). After the freeze-drying process, the experimental material was ground in a laboratory mill (A-11). Then, the ground samples were stored at −80 °C in small scyntylic tubes.

### 4.3. Chemical Analysis

#### 4.3.1. Dry Matter Analysis

The dry matter content of the *Aesculus* flowers and leaves was measured before the freeze-drying process. The dry matter content was determined using a scale as described by Polish Norm PN-R-04013:1988 [38]. Flower samples were dried at 105 °C for 48 h using an FP-25 W Farma Play dryer (Bytom, Poland). The dry matter content was calculated for the *Robinia* flower samples based on their mass differences and is given in units of g/100 g FW (fresh weight).

#### 4.3.2. Polyphenols—Extraction Parameters

Polyphenols (flavonols and phenolic acids) were measured using HPLC [39]. One hundred milligrams of freeze-dried flower or leaf powder was mixed with 5 mL of 80% methanol (HPLC grade) and mixed on a Vortex 326 M (Marki, Poland). Then, all samples were extracted in an ultrasonic bath (10 min, 30 °C, 5500 Hz). After 10 min of extraction, the flower and leaf samples were moved to a centrifuge (10 min, 6000 rpm, 5 °C). After centrifugation, each supernatant was collected in a clean Eppendorf tube and centrifuged again (5 min, 12,000 rpm, 0 °C). A total of 500 μL of supernatant was transferred to HPLC vials and analysed. Polyphenols (anthocyanins) were measured by HPLC [19]. The samples were extracted with a mixture of methanol and ultrapure water (80:20). After the first centrifugation (see previous section), 2.5 mL of the supernatant was collected into a new plastic tube, and then 2.5 mL of 10 M hydrochloric acid (HCl) and 5 mL of pure methanol were added. The samples were gently shaken and placed in a refrigerator (5 °C, 10 min). Then, 1 mL of each extract was transferred to an HPLC vial and analysed.

#### 4.3.3. Polyphenols—Equipment Description

For polyphenol separation and identification, a Synergi Fusion-RP 80i Column 250 × 4.60 mm (Phenomenex, Warsaw, Poland) was used. Analysis was conducted with the use of Shimadzu equipment (Chicago, IL, USA): two LC-20AD pumps, a CBM-20A controller, an SIL-20AC column oven, and a UV/Vis SPD-20 AV spectrometer. For separation of phenolic compounds (flavonols and phenolic acids), gradient conditions with a flow rate of 1 mL/min were used. Two gradient phases were used: 10% (*v*:*v*) acetonitrile and ultrapure water (phase A) and 55% (*v*:*v*) acetonitrile and ultrapure water (phase B). The phases were acidified with ortho-phosphoric acid (pH 3.0). The total time of the analysis was 38 min. The phase-time program was as follows: 1.00–22.99 min, 95% phase A and 5% phase B; 23.00–27.99 min, 50% phase A and 50% phase B; 28.00–28.99 min, 80% phase A and 20% phase B; 29.00–38.00 min, 95% phase A and 5% phase B. The wavelengths of detection were 250 nm for phenolic acids and 370 nm for flavonols.

Anthocyanins were separated under isocratic conditions with a flow rate of 1.5 mL/min. One mobile phase was used containing: acetic acid (5%), methanol (HPLC pure), and acetonitrile (HPLC pure) (70:10:20). The analysis time was 10 min with detection at 570 nm. The anthocyanins were identified by using 99.9% pure standards (Sigma-Aldrich, Warsaw, Poland) and the analysis times for the standards [39].

#### 4.3.4. Polyphenols—Results Calculation

All polyphenols were identified by using pure standards (Sigma-Aldrich, Warsaw, Poland) and the retention times for the internal standards. Standard curves prepared for all phenolic compounds are presented in Figure 1 and Figure 2. Pure phenolic standards were used for standard solution preparation. From each standard solution, five injections were made. Each time, the chromatographic pick area was determined and calculated based on the standard solution concentration. From standard curves, a mathematical equation was prepared. On the basis of the dilution coefficient and equation, the concentration of individual compounds was calculated.

#### 4.3.5. Carotenoids and Chlorophylls—Extraction Parameters

Carotenoids and chlorophylls were measured by HPLC [40]. One hundred and fifty-five milligrams of freeze-dried flower or leaf powder was mixed with 5 mL of 100% acetone (HPLC grade) and mixed on a Vortex 326 M (Marki, Poland). Then, all samples were extracted in a cold ultrasonic bath (15 min, 0 °C, 5500 Hz). After 15 min of extraction, the flower and leaf samples were moved to a centrifuge (15 min, 5000 rpm, 0 °C). After centrifugation, each supernatant was collected in a clean Eppendorf tube and centrifuged again (5 min, 12,000 rpm, 0 °C). A total of 900 μL of supernatant was transferred to HPLC vials and analysed, using a column injection of 50 µL.

#### 4.3.6. Carotenoids—Equipment Description

For carotenoid and chlorophyll separation and identification, a Max-RP 80i Column 250 × 4.60 mm (Phenomenex, Warsaw, Poland) was used. Analysis was conducted with Shimadzu equipment, as described above. For separation of carotenoids and chlorophyll compounds, gradient conditions with a flow rate of 1 mL/min were used. Two gradient phases were used: acetonitrile with methanol 90:10 (phase A) and methanol with ethyl acetate (68:32) (phase B). The total time of the analysis was 25 min. The phase-time program was as follows: 1.00–14.99 min, 100% phase A, 15.00–22.99 min, 40% phase A and 60% phase B; 23.00–27.99 min, 100% phase B. The wavelengths for detection were 445 nm for xanthophylls and 450 nm for carotenes and chlorophylls [40].

#### 4.3.7. Carotenoids—Results Calculation

All carotenoids and chlorophylls were identified by using pure standards (Sigma-Aldrich, Warsaw, Poland) and the retention times for the internal standards. Pure carotenoid (lutein, zeaxanthin, beta-carotene) and chlorophyll (a and b) standards were used for standard solution preparation. From each standard solution, five injections were made. Each time, the chromatographic pick area was determined and calculated based on the standard solution concentration. From standard curves, a mathematical equation was prepared. On the basis of the dilution coefficient and equation, the concentration of individual compounds was calculated.

#### 4.3.8. Antioxidant Activity Measurement and Calculation

##### ABTS Reagent Preparation

Twenty millilitres of distilled water was added to 0.0265 g of potassium persulfate (K2S2O8). Five millilitres of distilled water followed by 5 mL of a previously prepared aqueous solution of potassium persulfate was added to 0.0384 g of ABTS·+ (2′2-azinebis-3-ethylbenzothiazolin-6-sulfonic acid) reagent. The solution was prepared a minimum of 12 h before the planned assay and stored in a dark place. A total of 250 milligrams of the freeze-dried plant material was weighed into a plastic tube with a cap (50 mL), and 25 mL of distilled water was added. It was placed onto a vortex shaker (LP shaker Vortex, Labo Plus, Warsaw, Poland) for 60 s at 2000 rpm for complete mixing. Subsequently, the sample was incubated in a shaker incubator (IKA KS 4000 Control, IKA, Staufen im Breisgau, Germany) for 60 min (temperature 30 °C, 6000 rpm). After incubation, the sample was again shaken on a vortex shaker for 60 s for complete mixing and then centrifuged (centrifuge, MPW-380 R, Warsaw, Poland) at 5 °C and 8000 rpm for 20 min. After centrifugation, the supernatant was used for determinations. In 10 mL glass tubes, test extract solution, measured with a predetermined dilution scheme (0.5–1.5 mL), was then added to 3.0 mL of ABTS^•+^ cationic solution in PBS (phosphate-buffered saline). Absorbance measurements were taken exactly 6 min after incubation at room temperature. Absorbance was measured at a wavelength λ = 734 nm using a spectrophotometer (Helios, Thermo Scientific, Warsaw, Poland). The obtained measurements were calculated using a special formula including the dilution factor. The results were expressed as mmol of TE (Trolox equivalents per 100 g FW (fresh weight of flowers and leaves)) [41].

### 4.4. Statistical Analysis

All results were statistically assessed. For experimental purposes, Statgraphics Centurion 15.2.11.0 software (StatPoint Technologies, Inc., Warrenton, VA, USA) was used. The statistical calculations were based on two-way analysis of variance with the use of Tukey’s test (*p* = 0.05). A lack of statistically significant differences between the examined groups is indicated by similar letters. The standard error (SE) is provided with each mean value reported in the tables.

## 5. Conclusions

The conducted experiment confirms that not only fruits and seeds but also the leaves and flowers of two species of *Aesculus* (horse chestnut and red horse chestnut) contain different qualities and quantities of bioactive compounds, such as carotenoids, chlorophylls, and polyphenols. Furthermore, a two-year-long experiment provided more accurate data regarding the specific profiles and concentrations of bioactive compounds that occurred in the flowers and leaves of both species. Our results bring important information to science regarding the chemical composition of a less well-known species, the red horse chestnut. Due to the high concentrations of various bioactive compounds that they contain, horse chestnut and red horse chestnut flowers and leaves offer high biological value as food additives with strong bioactive and antioxidant capacity.

## Figures and Tables

**Figure 1 molecules-27-02279-f001:**
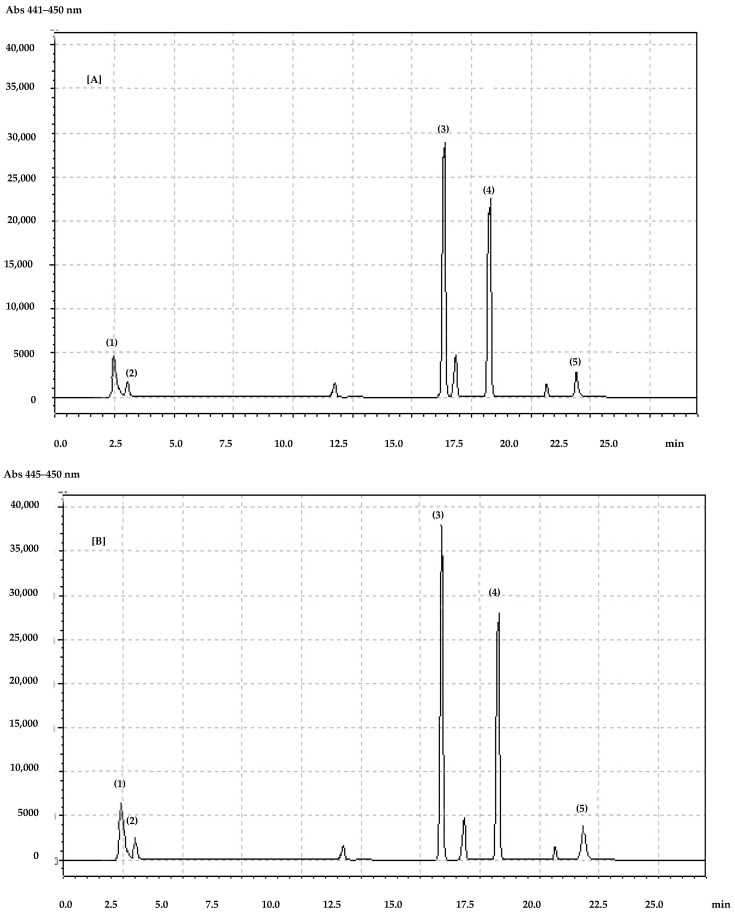
Chromatogram of carotenoids and chlorophylls identified in (**A**) horse chestnut and (**B**) red horse chestnut leaves. (1) lutein, (2) zeaxanthin, (3) chlorophyll b, (4) chlorophyll a, (5) beta-carotene.

**Figure 2 molecules-27-02279-f002:**
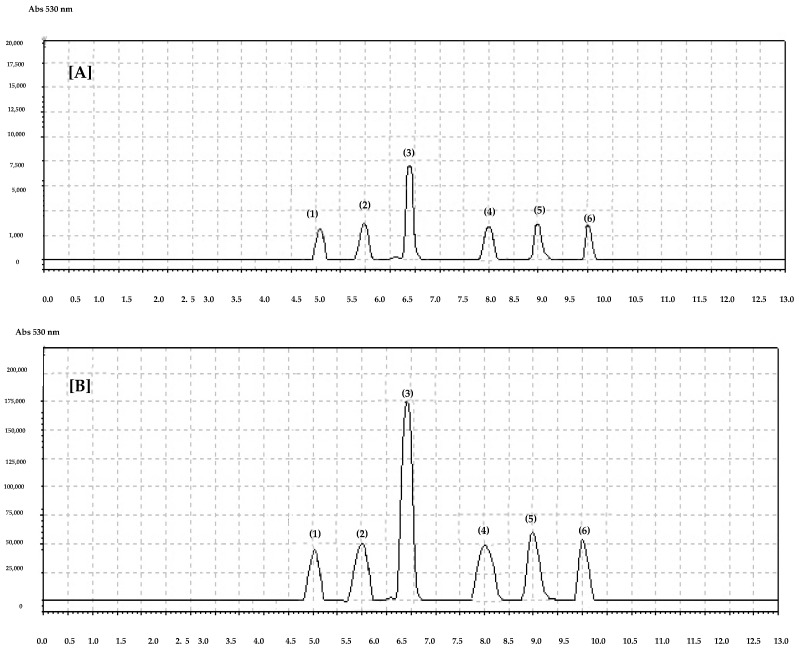
Chromatogram of anthocyanins identified in (**A**) horse chestnut and (**B**) red horse chestnut flowers. (1) cyanidin-3-O-glucoside, (2) pelargonidin-3-O-glucoside, (3) delphinidin-3-O-glucoside, (4) malvidin-3-O-glucoside, (5) peonidin-3-O-glucoside, (6) petunin-3-O-glucoside.

**Figure 3 molecules-27-02279-f003:**
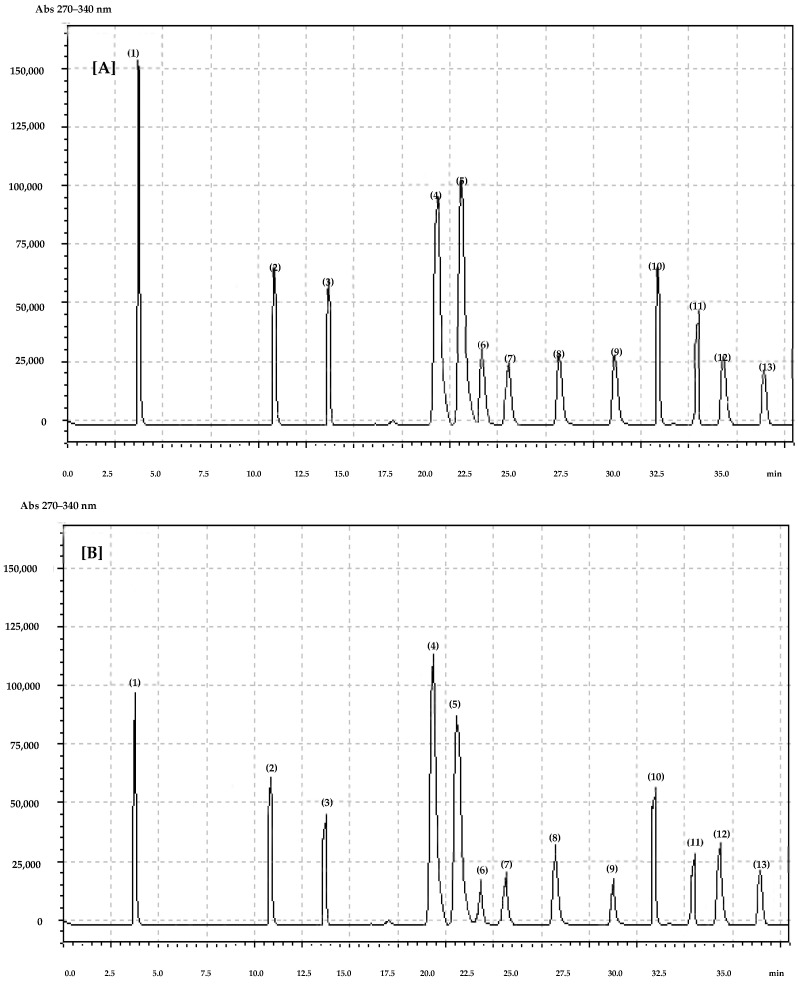
Chromatogram of polyphenols identified in (**A**) horse chestnut and (**B**) red horse chestnut flowers. (1) gallic acid, (2) chlorogenic acid, (3) caffeic acid, (4) quercetin-3-O-rutinoside, (5) p-coumaric, (6) ferulic acid, (7) kaempferol-3-O-glucoside, (8) myricetin, (9) luteolin, (10) quercetin, (11) quercetin-3-O-glucoside, (12) apigenin, (13) kaempferol.

**Figure 4 molecules-27-02279-f004:**
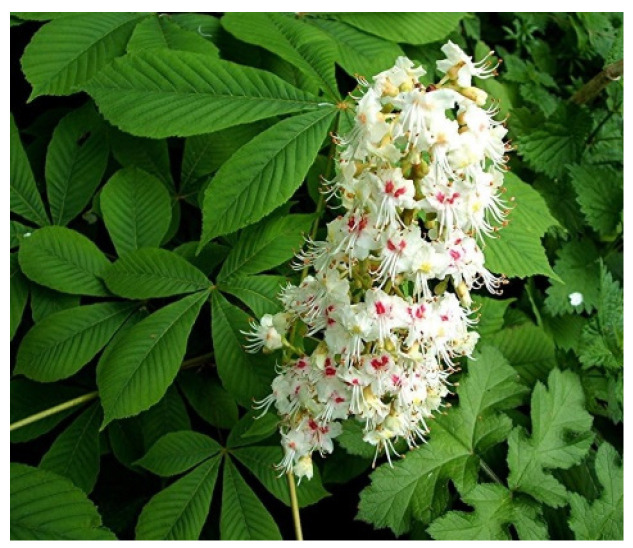
Flowers of *Aesculus hippocastanum* (horse chestnut flowers).

**Figure 5 molecules-27-02279-f005:**
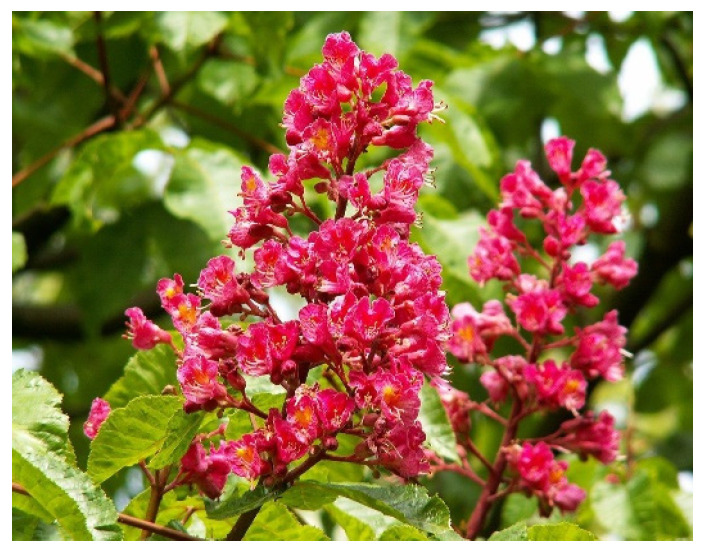
Flowers of *Aesculus* (×) *carnea* (red horse chestnut flowers).

**Table 1 molecules-27-02279-t001:** The content of bioactive compounds in total in flowers and leaves of horse chestnut and red horse chestnut (±standard error). N.S (statistically not significant).

**2018**	**Flowers**		**Leaves**	
	**Horse Chestnut**	**Red Horse Chestnut**	**Horse Chestnut**	**Red Horse Chestnut**
dry matter (g/100 g FW)	11.73 ± 0.27	11.01 ± 0.32	11.94 ± 0.16	13.83 ± 0.17
total carotenoids (µg/g FW)	36.55 ± 0.16	40.64 ± 0.27	750.27 ± 2.92	885.37 ± 4.29
total chlorophylls (µg/g FW)	18.95 ± 0.16	36.89 ± 0.28	1186.84 ± 13.76	1615.57 ± 12.96
total polyphenols (mg/g FW)	9.45 ± 0.05	8.25 ± 0.05	4.45 ± 0.03	5.72 ± 0.04
total phenolic acids (mg/g FW)	2.16 ± 0.05	0.35 ± 0.01	0.66 ± 0.01	0.46 ± 0.01
total flavonols (mg/g FW)	3.98 ± 0.04	2.49 ± 0.04	3.79 ± 0.02	5.27 ± 0.05
total anthocyanins (mg/g FW)	3.32 ± 0.04	5.41 ± 0.03		
**2019**	**Flowers**		**Leaves**	
	**Horse Chestnut**	**Red Horse Chestnut**	**Horse Chestnut**	**Red Horse Chestnut**
dry matter (g/100 g FW)	17.30 ± 0.70	15.30 ± 0.30	17.29 ± 0.43	18.29 ± 0.27
total carotenoids (µg/g FW)	38.69 ± 0.38	36.03 ± 0.07	697.55 ± 5.06	962.76 ± 4.59
total chlorophylls (µg/g FW)	25.88 ± 0.12	38.61 ± 0.18	1192.57 ± 13.69	1806.88 ± 15.39
total polyphenols (mg/g FW)	8.68 ± 0.26	8.96 ± 0.17	15.97 ± 0.94	13.31 ± 0.09
total phenolic acids (mg/g FW)	2.14 ± 0.17	1.26 ± 0.01	10.86 ± 0.84	7.36 ± 0.02
total flavonols (mg/g FW)	3.75 ± 0.11	2.70 ± 0.16	5.11 ± 0.10	5.95 ± 0.07
total anthocyanins (mg/g FW)	2.80 ± 0.12	5.00 ± 0.06		
	** *p* ** **-Value**			
	**Species**	**Organs**	**Years**	
dry matter	N.S.	N.S.	N.S.	
total carotenoids	<0.0001	<0.0001	<0.0001	
total chlorophylls	<0.0001	<0.0001	<0.0001	
total polyphenols	N.S.	N.S.	N.S.	
total phenolic acids	N.S.	0.026	N.S.	
total flavonols	N.S.	<0.0001	<0.0001	
total anthocyanins	<0.0001	<0.0001	<0.0001	

**Table 2 molecules-27-02279-t002:** Content of identified carotenoids and chlorophylls (µg/g FW) in both species of horse chestnuts’ flowers and leaves (mean value ± standard error).

**2018**	**Flowers**		**Leaves**	
	**Horse Chestnut**	**Red Horse Chestnut**	**Horse Chestnut**	**Red Horse Chestnut**
lutein	9.32 ± 0.03	10.73 ± 0.06	4.44 ± 0.09	9.71 ± 0.05
zeaxanthin	12.58 ± 0.12	14.44 ± 0.10	22.55 ± 0.04	31.60 ± 0.04
beta-carotene	14.65 ± 0.08	15.47 ± 0.15	723.27 ± 2.93	844.05 ± 4.33
chlorophyll a	7.46 ± 0.16	8.51 ± 0.16	616.01 ± 7.46	765.54 ± 12.43
chlorophyll b	11.46 ± 0.15	28.38 ± 0.12	570.83 ± 6.31	850.03 ± 3.34
**2019**	**Flowers**		**Leaves**	
	**Horse Chestnut**	**Red Horse Chestnut**	**Horse Chestnut**	**Red Horse Chestnut**
lutein	11.43 ± 0.60	11.49 ± 0.12	3.35 ± 0.07	8.68 ± 0.10
zeaxanthin	13.78 ± 0.07	11.19 ± 0.11	21.56 ± 0.08	30.38 ± 0.15
beta-carotene	13.48 ± 0.21	13.35 ± 0.14	672.63 ± 5.13	923.70 ± 4.76
chlorophyll a	6.32 ± 0.15	7.29 ± 0.07	556.86 ± 4.05	854.21 ± 13.26
chlorophyll b	19.56 ± 0.14	31.31 ± 0.11	635.71 ± 15.91	952.68 ± 2.63
	** *p* ** **-Value**			
	**Species**	**Organs**	**Years**	
lutein	<0.0001	<0.0001	<0.0001	
zeaxanthin	<0.0001	<0.0001	<0.0001	
beta-carotene	<0.0001	<0.0001	<0.0001	
chlorophyll a	<0.0001	<0.0001	<0.0001	
chlorophyll b	<0.0001	<0.0001	<0.0001	

**Table 3 molecules-27-02279-t003:** Content of identified anthocyanins (mg/g FW) in both species of horse chestnuts’ flowers (mean value ± standard error). N.S (statistically not significant).

**Flowers**	**2018**		**2019**	
	**Horse Chestnut**	**Red Horse Chestnut**	**Horse Chestnut**	**Red Horse Chestnut**
cyanidin-3-O-glucoside	0.19 ± 0.01	0.46 ± 0.01	0.14 ± 0.01	0.36 ± 0.01
pelargonidin-3-O-glucoside	1.55 ± 0.01	2.92 ± 0.02	1.34 ± 0.05	2.81 ± 0.06
delphinidin-3-O-glucoside	0.13 ± 0.01	0.18 ± 0.01	0.21 ± 0.01	0.15 ± 0.01
malvidin-3-O-glucoside	0.54 ± 0.02	0.25 ± 0.01	0.38 ± 0.03	0.26 ± 0.01
peonidin-3-O-glucoside	0.67 ± 0.01	0.85 ± 0.02	0.40 ± 0.02	0.77 ± 0.01
petunin-3-O-glucoside	0.24 ± 0.01	0.75 ± 0.01	0.33 ± 0.01	0.65 ± 0.01
	** *p* ** **-Value**			
	**Species**	**Years**		
cyanidin-3-O-glucoside	<0.0001	<0.0001		
pelargonidin-3-O-glucoside	<0.0001	<0.0001		
delphinidin-3-O-glucoside	N.S.	N.S.		
malvidin-3-O-glucoside	0.0061	0.0061		
peonidin-3-O-glucoside	0.0004	0.0004		
petunin-3-O-glucoside	<0.0001	<0.0001		

**Table 4 molecules-27-02279-t004:** Content of identified phenolic acids and flavonoids (mg/g FW) and antioxidant activity (mMol/100 g FW) in both species of horse chestnuts’ flowers and leaves (mean value ± standard error). N.S (statistically not significant).

**2018**	**Flowers**		**Leaves**	
	**Horse Chestnut**	**Red Horse Chestnut**	**Horse Chestnut**	**Red Horse Chestnut**
gallic	1.43 ± 0.04	0.31 ± 0.01	11.60 ± 0.12	8.35 ± 0.05
chlorogenic	0.24 ± 0.01	0.11 ± 0.01	0.14 ± 0.01	0.15 ± 0.01
caffeic	0.12 ± 0.01	0.16 ± 0.01	0.19 ± 0.01	0.06 ± 0.01
p-coumaric	0.68 ± 0.01	0.20 ± 0.01	0.46 ± 0.01	0.64 ± 0.01
ferulic	0.63 ± 0.01	0.10 ± 0.01	0.36 ± 0.02	0.08 ± 0.01
quercetin-3-O-rutinoside	0.75 ± 0.01	0.85 ± 0.01	1.23 ± 0.01	1.57 ± 0.01
kaempferol-3-O-glucoside	0.70 ± 0.02	0.64 ± 0.02	0.11 ± 0.01	0.78 ± 0.01
quercetin-3-O-glucoside	0.54 ± 0.01	0.23 ± 0.01	0.16 ± 0.01	0.36 ± 0.01
myricetin	0.19 ± 0.01	0.25 ± 0.01	0.35 ± 0.01	0.55 ± 0.02
luteolin	1.56 ± 0.01	0.25 ± 0.01	1.26 ± 0.01	1.66 ± 0.01
quercetin	0.11 ± 0.01	0.11 ± 0.01	0.12 ± 0.01	0.16 ± 0.01
kaempferol	0.12 ± 0.01	0.14 ± 0.01	0.56 ± 0.01	0.18 ± 0.01
ABTS	24.43 ± 0.07	29.57 ± 0.10	102.82 ± 0.21	115.73 ± 0.22
**2019**	**Flowers**		**Leaves**	
	**Horse Chestnut**	**Red Horse Chestnut**	**Horse Chestnut**	**Red Horse Chestnut**
gallic	1.46 ± 0.15	0.98 ± 0.01	10.15 ± 0.08	7.11 ± 0.04
chlorogenic	0.13 ± 0.01	0.11 ± 0.01	0.23 ± 0.01	0.17 ± 0.01
caffeic	0.17 ± 0.01	0.11 ± 0.01	0.16 ± 0.01	0.03 ± 0.01
p-coumaric	0.77 ± 0.01	0.35 ± 0.01	0.54 ± 0.01	0.76 ± 0.01
ferulic	0.55 ± 0.01	0.07 ± 0.01	0.22 ± 0.01	0.06 ± 0.01
quercetin-3-O-rutinoside	0.87 ± 0.01	0.95 ± 0.01	2.35 ± 0.01	2.54 ± 0.08
kaempferol-3-O-glucoside	0.86 ± 0.01	0.78 ± 0.01	0.13 ± 0.01	0.87 ± 0.01
quercetin-3-O-glucoside	0.42 ± 0.01	0.13 ± 0.01	0.13 ± 0.01	0.24 ± 0.01
myricetin	0.16 ± 0.01	0.18 ± 0.01	0.49 ± 0.02	0.45 ± 0.02
luteolin	1.19 ± 0.03	0.45 ± 0.22	1.03 ± 0.05	1.57 ± 0.01
quercetin	0.14 ± 0.01	0.09 ± 0.01	0.19 ± 0.01	0.16 ± 0.01
kaempferol	0.10 ± 0.01	0.12 ± 0.01	0.79 ± 0.04	0.12 ± 0.01
ABTS	28.40 ± 0.06	26.59 ± 0.12	98.41 ± 0.11	125.64 ± 0.29
	** *p* ** **-Value**			
	**Species**	**Organs**	**Years**	
gallic	<0.0001	<0.0001	0.0001	
chlorogenic	0.0072	N.S.	N.S.	
caffeic	<0.0001	0.0018	<0.0001	
p-coumaric	0.0001	0.0011	<0.0001	
ferulic	<0.0001	<0.0001	<0.0001	
quercetin-3-O-rutinoside	N.S.	<0.0001	N.S.	
kaempferol-3-O-glucoside	<0.0001	<0.0001	<0.0001	
quercetin-3-O-glucoside	0.0052	0.0002	<0.0001	
myricetin	0.0158	<0.0001	N.S.	
luteolin	0.0037	<0.0001	<0.0001	
quercetin	N.S.	0.0003	N.S.	
kaempferol	<0.0001	<0.0001	<0.0001	
ABTS	<0.0001	<0.0001	<0.0001	

## Data Availability

Not applicable.

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
