# Peer review of "Red Horse Chestnut and Horse Chestnut Flowers and Leaves: A Potential and Powerful Source of Polyphenols with High Antioxidant Capacity"

_molecules, 2022, doi:10.3390/molecules27072279_

Round 1
Reviewer 1 Report
Reviewer's Comments
Title: Red horse-chestnut and horse-chestnut flowers and leaves as potential powerful source of pro-healthy polyphenols with high antioxidant capacity- You can modify the title with the correct meaning. for Ex: Red horse-chestnut and horse-chestnut flowers and leaves: a potential and powerful source of polyphenols with high antioxidant capacity
- rewrite the abstract- includ the polyphenol content values and antioxidant values.
- methods are well described and used international standard.
- Results are very attractive and the discussion part is well written.
- Improve the conclusion
- Edit the reference list as per journal guidelines
Author Response
Reviewer no. 1
Thank you very much for the review and for your positive recommendation to publish our manuscript in the ‘Molecules’ journal. Below you can find answers for your comments and suggestions:
Comment 1: “You can modify the title with the correct meaning. for Ex: Red horse-chestnut and horse-chestnut flowers and leaves: a potential and powerful source of polyphenols with high antioxidant capacity”
Authors’ response: Thank you for pointe better formation of manuscript title. According to Reviewer suggestion, the tittle has been change.
Comment 2: “rewrite the abstract- includ the polyphenol content values and antioxidant values.”
Authors’ response: According to Reviewer suggestion missing compounds values were insert to abstract section
Comment 3: “Improve the conclusion”
Authors’ response: According to Reviewer suggestion sub-section conclusions is slightly correct by substantively and language ways. Language certificate was attached to not-publish materials in time of corrected manuscript submitting.
Comment 4: “Edit the reference list as per journal guidelines”
Authors’ response: According to Reviewer suggestion, all positions from reference list were carefully checked and correct according Molecules
Reviewer 2 Report
The manuscript entitled "Red horse-chestnut and horse-chestnut flowers and leaves as potential powerful source of pro-healthy polyphenols with high antioxidant capacity" and signed by Agnieszka Monika Bielarska et al. contains phytochemical analysis data of two chestnuts trees for leaves and flowers. Although the novelty is quite modest, the study is well organized. Some important questions indicated are still open and need to be answered before acceptance. Other issues need to be clarifies as follows:
1. Abstract:
- Half of the abstract contains description of the scientific context that should be rather mentioned in the Introduction. A short one-sentence context description is enough in the abstract, the rest should be rather presenting most important findings, results and implications. Aims of the study is also welcome and it is mentioned in this abstract.
- Analytical technique that was employed in the study must be mentioned in the abstract
- Line 21, the name of the hybrid is written wrong "Aesculus x carenea", instead of x letter, the authors have to use a multiplication sign (×). This correction has to be done throughout the text as well.
2. Introduction:
- It presents very well the scientific context with mentions about the polyphenols implications in these tree species. However, there is space for information regarding other known papers that studied chestnuts trees phytochemical composition of different parts of the plant. The introduction mentions very little about it, although there are lots of literature available.
3. Results:
- Table 1: when standard error (SE) is indicated, n should be stated (number of experimental and/or technical replicates. SE values of 0.0 are not common to be indicated, how could be the error 0? Probably its is 0.02 or 0.003 and was down rounded. Errors are never down rounded or indicated with 0 values. Values with two decimals should list error with 2 decimals, not one.
- It is unclear why the y-axis of the chromatograms has a "mV" unit of measurement since in the Experimental section a UV-vis detection and a wavelength is mentioned. Whys is that? On the y-axis the Abs and the wavelength should be indicated.
- The authors indicate that they both dried (at too high temperature, 105 C, in my opinion) and freezed-dried the plant material. For what types of procedure to the Tables indicated the values?
- The shape of the chromatographic peaks in Figure 2 is very strange, for compound 4, in A. it is rather triangular while in B its kurtosis is much higher. Why is that? The elution time for some of the compounds is also quite different, why?
4. Discussions:
- The first sentence of this section "In the present study, species with white flower colour also contained more dry matter" is not clear. More than what?
- Why do the authors repeat the indications of the results in the discussion section? Weren't they indicated in the Results section? Then, why not making a Results and Discussions section?
5. Materials and Methods:
- The section is very well written, with important and relevant details indicated.
- Photo 1 and photo 2 are mentioned but they are not displayed in the manuscript.
- Drying the vegetable material at 105 C in't too high and induce chemical modifications of the polyphenols? Especially for 48 hours.
- line 331, remove the dash from the "4.3.8-". In this section the wavelength is always indicated with lambda Greek letter, not alpha, please correct. There is no indication about the protocol of the preparation of the cationic radical ABTS•+.
6. Conclusions: concise and supported by the results.
Author Response
Reviewer no. 2
Thank you very much for the review and for your positive recommendation to publish our manuscript in the ‘Molecules’ journal. Below you can find answers for your comments and suggestions:
The manuscript entitled "Red horse-chestnut and horse-chestnut flowers and leaves as potential powerful source of pro-healthy polyphenols with high antioxidant capacity" and signed by Agnieszka Monika Bielarska et al. contains phytochemical analysis data of two chestnuts trees for leaves and flowers. Although the novelty is quite modest, the study is well organized. Some important questions indicated are still open and need to be answered before acceptance. Other issues need to be clarifies as follows:
- Abstract:
Comment 1: “ Half of the abstract contains description of the scientific context that should be rather mentioned in the Introduction. A short one-sentence context description is enough in the abstract, the rest should be rather presenting most important findings, results and implications. Aims of the study is also welcome and it is mentioned in this abstract.”
Authors’ response: Authors agree with Reviewer suggestion. Only one short sentence was left in Abstract about background of experiment. The rest of redundant considerations were removed from this part of manuscript. The aim of the study is now in Abstract part.
Comment 2: “ Analytical technique that was employed in the study must be mentioned in the abstract”
Authors’ response: According to Reviewer suggestion short mention of analytical technique was added to Abstract section.
Comment 3: “ Line 21, the name of the hybrid is written wrong "Aesculus x carenea", instead of x letter, the authors have to use a multiplication sign (×). This correction has to be done throughout the text as well.”
Authors’ response: According to Reviewer suggestion the correction of the misspelled hybrid mark has been introduced throughout the manuscript.
- Introduction:
Comment 4: “ It presents very well the scientific context with mentions about the polyphenols implications in these tree species. However, there is space for information regarding other known papers that studied chestnuts trees phytochemical composition of different parts of the plant. The introduction mentions very little about it, although there are lots of literature available”.
Authors’ response: According to Reviewer suggestion more information about concentration of polyphenols in different horse chestnuts botanical part (leaves, flowers, seeds and bark) were added to introduction section. Information are obtained from scientific articles and appropriate references were added to reference list.
- Results:
Comment 5: “ Table 1: when standard error (SE) is indicated, n should be stated (number of experimental and/or technical replicates. SE values of 0.0 are not common to be indicated, how could be the error 0? Probably its is 0.02 or 0.003 and was down rounded. Errors are never down rounded or indicated with 0 values. Values with two decimals should list error with 2 decimals, not one”
Authors’ response: Authors agree with Reviewer suggestion and want to apologize. This situation resulted from rounding the value to one decimal place. At present, all mean values ​​and the corresponding standard error values ​​have been carefully checked. As suggested by the Reviewer, all values ​​are given with accuracy to two decimals.
Comment 6: “It is unclear why the y-axis of the chromatograms has a "mV" unit of measurement since in the Experimental section a UV-vis detection and a wavelength is mentioned. Whys is that? On the y-axis the Abs and the wavelength should be indicated”
Authors’ response: Authors agree with Reviewer suggestion and want to apologize for a mistake. Of course according to described methodology procedure, on the y-axis should be absorbantion (Abs.) as well together with their value or value range. This mistake were corrected on the all figures presenting chromatograms.
Comment 7: “ The authors indicate that they both dried (at too high temperature, 105oC, in my opinion) and freezed-dried the plant material. For what types of procedure to the Tables indicated the values?
Authors’ response: Authors want to clarify two ranges of drying temperatures used in the experiment. The procedure of drying flowers and leaves at a temperature of 105oC was carried out only to determine the dry weight in the examined parts of the plants. It is a procedure approved by the Polish Committee for Standardization and Quality. Everything is in accordance with the issued and quoted Quality Standard. All measurements of bioactive compounds (polyphenols, carotenoids and chlorophylls) were carried out with freeze-dried plant material. After freeze drying process plant material was grounded and keep in -80oC to avoid losing decomposition of bioactive compounds. According to my knowledge in time of freeze-drying as well in plant powder there is no decomposition of bioactive substances. To summarized: in Tables drying in +105oC only for dry matter determination, frieze-drying (-40oC and 0.100 mBa) for bioactive compounds determination.
Comment 8: “The shape of the chromatographic peaks in Figure 2 is very strange, for compound 4, in A. it is rather triangular while in B its kurtosis is much higher. Why is that? The elution time for some of the compounds is also quite different, why?..”
Authors’ response: Authors agree with Reviewer suggestion and want to apologize. Strange shape of picks is results technical problems with trying to change the width and length of a photo (not an object). It was a technical problem. Now there are chromatograms in a different form and are more distinctive. The peaks are not triangular but typical.
- Discussions:
Comment 9: “The first sentence of this section "In the present study, species with white flower colour also contained more dry matter" is not clear. More than what?”
Authors’ response: Authors want to apologize for mistake. Frankly speaking the sentence was not finish and now is corrected.
Comment 10: “Why do the authors repeat the indications of the results in the discussion section? Weren't they indicated in the Results section? Then, why not making a Results and Discussions section?”
Authors’ response: Authors want to explain such situation. According to Authors guidelines both section Results and Discussion should be presented separately. Even in the used one template there was such a information. In sub section Results Tables with values are presented as well they description (without values in manuscript text). While in Discussion section only few times values from previous section are cited to make reading more comfortable.
- Materials and Methods:
The section is very well written, with important and relevant details indicated.
Comment 11: “Photo 1 and photo 2 are mentioned but they are not displayed in the manuscript”
Authors’ response: Authors want to apologize for mistake. Both Photo 1 and 2 were loaded as separate files, not as paste with manuscript. Now mistake is corrected. Photo 1 and 2 are in manuscript text.
Comment 12: “Drying the vegetable material at 105 C in't too high and induce chemical modifications of the polyphenols? Especially for 48 hours.”
Authors’ response: Please see reply for comment no. 7
Comment 13: “line 331, remove the dash from the "4.3.8-".”
Authors’ response: According to Reviewer suggestion the dash from the “4.3.8. was removed.
Comment 14: “In this section the wavelength is always indicated with lambda Greek letter, not alpha, please correct.”
Authors’ response: According to Reviewer suggestion wrong letter “alpha” (α) was replaced by correct letter lambda (λ) in manuscript text.
Comment 15: There is no indication about the protocol of the preparation of the cationic radical ABTS•+.
Authors’ response: According to Reviewer suggestion part of protocol about preparation of the cationic radical ABTS•+. was added to manuscript text.
- Conclusions: concise and supported by the results.
Round 2
Reviewer 2 Report
The revised manuscript could be accepted for publication.